# Purchasing Bakery Goods during COVID-19: A Mind Genomics Cartography of Hungarian Consumers

**Barbara Biró** and **Attila Gere** *

Department of Postharvest, Supply Chain, Commerce and Sensory Science, Institute of Food Science and Technology, Hungarian University of Agriculture and Life Sciences, H-1118 Budapest, Hungary; barbarabirophd@gmail.com
* Correspondence: gereattilaphd@gmail.com

**Abstract:** At both global and national levels, COVID-19 caused huge changes both in politics and economics, including the agricultural sector and the food industry, from producers, manufacturers, and traders to consumers. Since March 2020, many restrictions and protective measures were introduced worldwide, which only began to be relaxed in the last weeks of spring 2021 as the number of people vaccinated against the coronavirus increased in Hungary. The aim of this study was to investigate the attitudes of Hungarian consumers toward food purchases during the COVID-19 pandemic, in terms of safety. The research was based on the purchase of bakery products, which are basic food products and are most often found in an unpackaged form in Hungarian stores. The BimiLeap® study, a revolutionary tool for uncovering people's minds, was completed by 125 participants, gathered by a snowballing technique. There were no significant differences among consumers' attitudes based on the traditional socio-demographic descriptors; however, the mindset-based classification was able to differentiate significantly. The three identified mindsets covered people who themselves consider bakery products, the purchase method, and being in the store as the highest risk of a potential COVID infection.

**Keywords:** Mind Genomics; mindsets; COVID-19; bakery products; food purchase

## 1. Introduction

The pandemic caused by a new type of coronavirus (SARS-CoV2) has reshaped the world. At both global and national levels, it has caused huge changes both in politics and economics, including the agricultural sector and the food industry, from producers, manufacturers, and traders to consumers. It is questionable which of the changes resulting from the disease control measures introduced will be permanent and to what extent.

In Hungary, a state of emergency was declared by the Government on 11 March 2020, which led to the introduction of a special legal order [1]. This legal order has already been extended several times and, at the time of writing this article, under the Act XL of 2021, the Act I of 2021 about the special legal order will expire on the 15th day after the first day of the autumn session of Parliament [2]. Over the past year, several restrictions and protective measures have been introduced, such as mandatory wearing of masks in public places, shops, and public transport; curfews; and the limitation of the number of customers in shops. In higher education, online education started as soon as the emergency was declared in 2020, and primary and secondary schools also switched to online education at the end of last year. Travel restrictions were also implemented, for both Hungarian and foreign citizens [3]. After 30 April 2021, vaccinated citizens are allowed to enter indoor restaurants and entertainment areas, using their vaccination certificate card to enter [4]. As the number of people vaccinated with at least the first dose exceeded 5.5 million, almost all restrictions have been lifted, including the mandatory mask wearing in stores, public transport, and public places [5].

Despite the fact that the restrictions and protective measures were based on the recommendations by many national and international professional organizations (e.g., World Health Organization (WHO)) and have proven to be effective in other countries, there is skepticism about them among some groups of Hungarian citizens and there is also a doubt about the vaccines used. Several doctors and health professionals who call themselves "epidemic critics" have spoken out against the restrictions and vaccination. They also set up protest pages and groups on social media, the most popular of which had over 100,000 followers at the time of its shutdown [6]. At the end of February, a demonstration was also organized against the restrictions and the vaccination [7]. Currently, the main issues among these groups are the disadvantages of not having a document proving immunity (vaccination certificate card), questioning the effectiveness of vaccines, and spreading fake news about their serious side effects.

The pandemic and the restrictions affected every aspect of everyday life, including shopping. When the restrictions and the protective measures were introduced, consumers started panic buying and stockpiling, causing shortages of many items such as flour, sugar, and fresh meat. Customers preferred to visit smaller stores rather than larger super- and hypermarkets and preferred to pay by card rather than cash [8]. The least purchased products were clothing, handcrafted products, and unpacked bakery products [9]. Bread consumption in Hungary had already fallen from 44.5 kg/capita to 34.4 kg between 2010 and 2018 [10], while the popularity of white bread also decreased, from 76% to 61% between 2007 and 2017 [11]. It is questionable whether the pandemic has exacerbated these downward trends, as home bread and pastry making has become very popular while the number of purchases of bakery products has decreased. Based on the work of Sikos and co-workers, this may be due to the abstention from unpackaged food products [9]. Despite the downward trend, this is still a large quantity, so the quality of bread and other bakery products available in stores is a major issue.

In Hungary, bread is most often found in stores in its unpackaged form, which, together with other unpackaged food products, raises food safety and quality issues. As ready-to-eat products, bread and bakery goods are usually consumed without reheating. Packaged breads have been proven to remain microbiologically safe and retain their desired sensory properties for longer, thus extending their shelf life [12]. Properly handled bread and bakery goods do not pose a risk due to the high baking temperature and the adequate moisture content, but the potential for post-process surface contamination is high. Due to inadequate storage conditions and hygiene, products can be contaminated with various mold species (e.g., *Rhizopus* sp., *Penicillium* sp.) and bacteria (e.g., *Bacillus subtilis*, *Bacillus licheniformis*), which can cause foodborne illnesses [13]. For other foods, ready-to-eat salads sold on self-service counters and fresh, unpacked fruit and vegetables can pose a food safety risk due to their microbial load (e.g., *Escherichia coli*, *Bacillus cereus*, *Clostridium perfringens*). To prevent these illnesses, it is advisable to educate both store staff and consumers about the proper storing and handling of these products, and to apply controlling and monitoring systems in the stores. [14,15]. Despite the fact that the possibility of spreading the new type of coronavirus through food is not significant [16], increased attention must be paid to hygiene standards.

Uncovering people's minds about different topics has always been a critical question for researchers. Different surveys [17], focus group interviews [18], or ConJoint analyses [19] are available to complete such studies. Among these many options, the ConJoint-based Mind Genomics showed an increasing trend and success in the past decades thanks to its flexibility and wide range of applications. Mind Genomics was developed by Dr. Howard Moskowitz and introduced in 2006 [20]. Similar to ConJoint analysis, it requires a topic, a set of questions coupled with possible answers to create hypothetical products, services, or short stories, which are rated on a predefined rating scale. It has been applied in various fields, such as uncovering the mind of consumers regarding meat analogues [21], people's reaction to COVID-19 restrictions [22], insect-based foods, [23] or consumer perception of health loss [24]. In the past few years, Mind Genomics followed

international trends and has been transferred to mobile applications under the name of BimiLeap®. Mind Genomics (and therefore BimiLeap®) shares some similarities with Con-Joint analysis, except that BimiLeap® creates a special experimental design that presents 24 unique vignettes to each participant in an online, platform-independent way.

As mentioned earlier, unpackaged bakery products present some questions regarding the spread of COVID-19 or other infections. Therefore, we aimed to map consumers' attitudes towards purchasing food products during the pandemic using BimiLeap®. Through BimiLeap® we aimed to identify different mindsets based on predefined elements answering the questions where and how to purchase different types of bakery products under different restrictions and protective measures.

## 2. Materials and Methods

### 2.1. BimiLeap®

In order to uncover the mind of participants, the advanced version of Mind Genomics, BimiLeap®, was used. BimiLeap® is freely available to create, run, analyze, and disseminate Mind Genomics studies. [23,25].

The structure of creating a BimiLeap® study is straightforward. In the first step, the researcher needs to define the topic in question, which in our case was the understanding of people's bakery purchasing behaviors during the pandemic. In the next step, four questions or silos need to be defined, covering the topic as completely as possible to receive the desired information. These questions are then filled with four elements each, providing different answers to the questions. Using the elements, BimiLeap® creates so-called vignettes, which will be later evaluated by the participants. Vignettes are combined using strictly one element from each silo; however, not all silos are used in order to create a balanced presentation of elements throughout the study. Vignettes, therefore, can be comprised of two to four elements. In the presented study, the questions and the elements were defined by a group of experts after a careful analysis of the bakery market of Hungary and existing COVID-19 relevant literature (e.g., newspaper articles, Magyar Közlöny, recommendations of the National Public Health Center (NPCH) and the World Health Organization (WHO). The created BimiLeap® study is presented in Table 1.

**Table 1.** Silos (questions) and elements (answers) used in the BimiLeap® study.

| | **Question A: Method of Purchase** |
|---|---|
| A1 | I go to the store myself, using public transport |
| A2 | I go to the store myself on foot/by car |
| A3 | I order online |
| A4 | Brought by a friend/family member |
| | **Question B: Places of purchase** |
| B1 | Hypermarket |
| B2 | Supermarket |
| B3 | Bakery |
| B4 | Convenience store |
| | **Question C: Packaging** |
| C1 | Product without packaging |
| C2 | Product packaged in the store |
| C3 | Pre-packaged product |
| C4 | Frozen product |
| | **Question D: Protective measures** |
| D1 | No protective measures in the store |
| D2 | Mandatory wearing of masks and distance keeping in the store |
| D3 | Mandatory disinfection of the hands at the arrival at the store |
| D4 | Limitation of the number of people in the store |

Supermarkets: located in both residential and city center areas; floor area: 400–2500 m$^2$; number of stock-keeping units (SKUs): thousands; number of cash registers: between 3–10 units. Hypermarkets: located on the outskirts of cities; floor area: over 2500 m$^2$; number of SKUs: more than 10,000; number of cash registers: more than 10.

Respondents are then asked to rate the created vignettes on a predefined 9-point scale. In the presented study, the following rating question was used, "*How safe do you feel the presented bakery-purchasing situation?*", and participants could provide their answers from 1 (not safe at all) to 9 (safe). Each respondent sees and rates 24 different vignettes. So, with 100 people, the system creates 100 × 24 or 2400 DIFFERENT vignettes. The structure of the vignettes (e.g., the presence and absence of each of the 16 answers for each vignette) is stored in a table that is available to download once the study is closed [26].

BimiLeap® includes further classification questions, such as age and gender (male, female, non-binary) as well as an additional question defined by the authors. In the current study, it regarded the place of residence (capital city, city, town, and village), at the beginning of the questionnaire. At the end of the study, a non-mandatory, open-ended question ("*What is your opinion on the restrictive/protective measures currently in place?*") was asked.

The finished study was shared online following the snowball method to collect as many participants as possible. An example vignette of the created BimiLeap® study is presented by Figure 1 (mobile view). Participants see the number of the vignette in the upper left corner, a short description of the task, the rating questions, and the elements that should be rated. At the bottom of the screen, the rating scale is presented with the two labels of the endpoints. As soon as the participant clicks on an answer, the next vignette is immediately presented and there is no possibility to turn back to any previous vignettes.

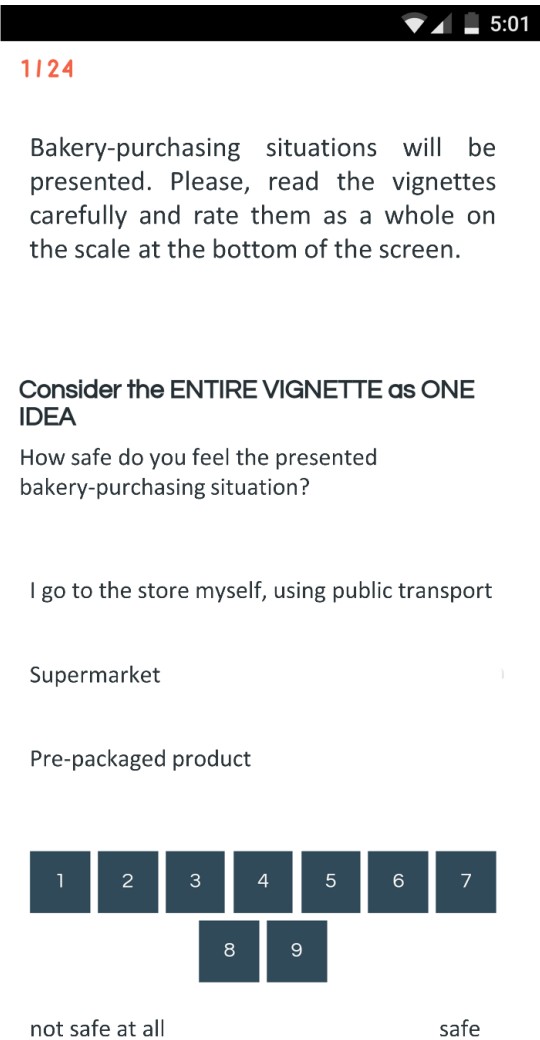

**Figure 1.** Example vignette used in the study.

*2.2. Participants*

The study was available between 6–26 April 2021 and altogether 125 participants successfully filled out the online test. In addition to these 125 participants, a further 72 opened but did not complete the study; therefore, their results were automatically removed. The age of participants ranged between 19 and 65 years with an average of $28.65 \pm 9.70$ years. Women were overrepresented in the study, as we aimed to choose respondents who do regular shopping of bakery products; therefore, 100 females and 25 males participated (80% vs. 20%, respectively). No participant chose the non-binary option. Participants lived in Budapest (capital of Hungary) (48%), in cities (16%), towns (21%), and villages (15%).

*2.3. Data Analysis*

BimiLeap$^{®}$ uses ordinary least squares (OLS) regression to assess the effect of the elements on the ratings of the participants. The input data matrix has the vignettes presented to the participants (24 per participant) in its rows and the elements (16 elements) in its columns plus the rating given by the participants. The variables are coded by 0's and 1's, depending on if the given element was presented or not in the given vignette. The given rating is also registered and stored in the variable "Rating". In order to identify the most influencing variables, Rating is transformed by BimiLeap$^{®}$. First, ratings of 1–6 on the scale are transformed to 0 (i.e., low/weak feeling), while ratings of 7–9 on the scale are transformed to 100 (e.g., high/strong feeling), creating the so-called top analysis. Then, a small random number is added to generate small standard deviation. That lack of variation would cause the OLS to crash. The OLS is performed on these data, elements (A1-D4) are used as independent variables, and the binarized ratings are used as dependent variables (Table 2). In order to be able to evaluate the lower endpoint of the rating scale (not safe at all), a similar analysis is done on the bottom end of the scale. For these, BimiLeap$^{®}$ transforms ratings 1–3 to 100 and 4–9 to 0, creating the so-called bottom analysis. With this step, we can focus on those elements that generated the lowest ratings on the rating scale, e.g., "not safe at all".

**Table 2.** First six rows of the input matrix for OLS regression.

| A1 | A2 | A3 | A4 | B1 | B2 | B3 | B4 | C1 | C2 | C3 | C4 | D1 | D2 | D3 | D4 | Rating | Binarized Rating |
|---|---|---|---|---|---|---|---|---|---|---|---|---|---|---|---|---|---|
| 1 | 0 | 0 | 0 | 0 | 0 | 0 | 0 | 0 | 0 | 1 | 0 | 1 | 0 | 0 | 0 | 9 | 100.7249753 |
| 1 | 0 | 0 | 0 | 0 | 1 | 0 | 0 | 0 | 0 | 0 | 1 | 0 | 0 | 0 | 1 | 5 | 0.427754649 |
| 0 | 0 | 0 | 0 | 1 | 0 | 0 | 0 | 0 | 0 | 0 | 1 | 1 | 0 | 0 | 0 | 9 | 100.461673 |
| 0 | 0 | 1 | 0 | 0 | 0 | 1 | 0 | 0 | 1 | 0 | 0 | 0 | 0 | 0 | 0 | 7 | 100.1592801 |
| 1 | 0 | 0 | 0 | 0 | 0 | 1 | 0 | 0 | 0 | 0 | 0 | 0 | 0 | 1 | 0 | 5 | 0.692818016 |
| 0 | 0 | 0 | 0 | 1 | 0 | 0 | 0 | 0 | 1 | 0 | 0 | 0 | 0 | 0 | 0 | 6 | 0.5522144 |

OLS is run on an individual level, meaning that a separate OLS is run on each of the 24 rows of the input matrix, as each participant rated 24 vignettes. OLS generates regression coefficients, which are stored; therefore, each participant receives a vector of 17 values, one for the additive constant and 16 coefficients for the 16 elements (dependent variables). As regression coefficients describe the relationship between a predictor variable and the response, the extent of the obtained coefficients tells us the effect of the given element on the rating given by the participants.

Participants are then clustered using *k*-means clustering using these regression coefficients and Pearson 1-R distance to create similarly thinking clusters, the so-called mindsets [27]. As the mathematical formula tells us, BimiLeap$^{®}$ does not use a priori segmentation; rather, the pattern of the coefficients helps us to group respondents to create mindsets' thinking as similar as possible. By default, BimiLeap$^{®}$ clustered the respondents

into two mindsets and then into three mindsets, because they represent different patterns of thinking about the same topic. [28]. Although these analyses are done automatically by BimiLeap®, the presented study was re-analyzed manually to get full control over the data set. The applied clustering indices suggested that three mindsets should be kept. Data analysis was done using R-project (version R-3.6.0) and lm.beta package [29].

## 3. Results and Discussion

### 3.1. Results of the Total Panel, Gender, and Place of Residence

The questionnaire was completed after the tightening in March 2021 (curfew extension, closure of stores and schools), before the introduction of the relaxations predicted after reaching 2.5 million people vaccinated with at least the first dose. Table 3 shows the results of the total panel, and results disaggregated by gender and by type of residence.

**Table 3.** Regression coefficients for models relating the presence/absence of the elements to the rating of safety, after binary transformation. The highest coefficients of each group are colored as gray. High positive coefficients denote a strong feeling of safety, while low negative coefficients mean the opposite. Coefficients around 0 mean neutral feelings. Superscript letters denote the homogenous subsets defined by Tukey post hoc test ($p < 0.05$).

| Code | Elements | Total | Male | Female | Capital City | City | Town | Village |
|------|----------|-------|------|--------|--------------|------|------|---------|
| A1 | I go to the store myself, using public transport | −1 | −1 | −1 | −2 | −8 | 4 | 2 |
| A2 | I go to the store myself on foot/by car | 0 | 1 | 0 | 1 | −8 | 3 | −2 |
| A3 | I order online | 0 | 7 | −1 | 4 | −11 | 4 | −1 |
| A4 | Brought by a friend/family member | −1 | 3 | −2 | 3 | −14 | −8 | 8 |
| B1 | Hypermarket | 2 | 3 | 3 | 2 | −6 | 7 | 6 |
| B2 | Supermarket | 4 | 4 | 4 | 2 | 2 | 7 | 8 |
| B3 | Bakery | −1 | −6 [a] | 0 [b] | −1 | −6 | 1 | 2 |
| B4 | Convenience store | 1 | 0 | 1 | −2 | 4 | 3 | 3 |
| C1 | Product without packaging | 4 | 6 | 3 | 7 | 8 | −1 | −3 |
| C2 | Product packaged in the store | 6 | 9 | 5 | 8 | 6 | 2 | 2 |
| C3 | Pre-packaged product | 2 | 5 | 2 | 5 | 5 | 1 | −5 |
| C4 | Frozen product | −3 | 5 | −4 | 1 | 0 | −4 | −13 |
| D1 | No protective measures in the store | −4 | −6 | −3 | 2 | −6 | −11 | −8 |
| D2 | Mandatory wearing of masks and distance keeping in the store | −2 | 0 | −2 | 1 | −3 | 0 | −11 |
| D3 | Mandatory disinfection of the hands at the arrival at the store | −4 | −9 | −3 | −2 | −2 | −4 | −10 |
| D4 | Limitation of the number of people in the store | −3 | 0 | −4 | 0 | −10 | −4 | −3 |

Since the coefficients and their differences were too small, it was not possible to draw any firm conclusions from the results of the total panel. In general, locally packaged products were considered the safest. The respondents were skeptical about the restrictive measures, none of which were considered really safe. Of the types of stores, they were most confident in the safety of hypermarkets and supermarkets.

Comparing women and men, the coefficients showed that men perceive online shopping to be much safer than women, but among shopping locations, hyper- and supermarkets were considered to be equally safe. While men considered all types of packaging as safe, women were more doubtful about frozen products. The only significant difference

between men and women was found in the case of purchasing in a bakery, which was considered less safe by men.

Residents of the capital city considered online shopping to be the safest, and they trusted unpacked and in-store packaged bakery products the most. The latter was also true for people living in cities, but they thought that shopping in a small shop was the safest. Those living in towns thought that ordering online and using public transport to get to the shop were the safest and had the greatest trust in shopping in hypermarkets and supermarkets. People living in villages also thought that hypermarkets and supermarkets were the safest places to shop and having a friend or family member do the shopping and deliver the product to their homes was the safest.

Looking at the results of the total panel, respondents did not consider any of the listed protective measures to be safe and had less trust in bakeries and frozen products.

Men did not consider it safe to buy bakery products from a bakery, and did not consider the lack of protective measures or the mandatory disinfection of their hands to be safe. Women did not consider either measure to be particularly safe and, unlike men, they did not trust frozen products.

Respondents living in the capital were generally neutral about almost every element: The small coefficients suggest that they did not consider them safe or unsafe. City residents, on the other hand, did not consider any way of purchasing bakery products to be safe, especially if the products were taken home by a friend or family member. They also did not consider hypermarkets and bakeries to be safe, and they had a lack of trust about the limitation of the number of customers in stores. People living in towns also considered to be the least safe if they did not buy the product themselves, and if there were no protective measures in force. Villagers completely rejected frozen products and did not consider any of the protective measures to be safe.

### 3.2. Identifying Mindsets

The emergent mindsets showed three distinct groups (Table 4). Mindset 1 appeared to have more trust in supermarkets, mostly when they go shopping themselves, Mindset 2 appeared to have more trust in the protective measures in the stores and products packaged in-store, whereas Mindset 3 appeared to have more trust in every type of the products, especially if they order them online. The three mindsets showed completely different pictures about what they consider safe:

**Table 4.** Regression coefficients for the top scores of the three mindsets. High positive coefficients mean that respondents did feel the element safer. The highest coefficients of each group are colored as gray. Superscript letters denote the homogenous subsets defined by Tukey post hoc test ($p < 0.05$).

| Code | Elements | Mind-Set 1 ($n = 27$) | Mind-Set 2 ($n = 53$) | Mind-Set 3 ($n = 45$) |
|---|---|---|---|---|
| A1 | I go to the store myself, using public transport | 15 [b] | −15 [a] | 5 [b] |
| A2 | I go to the store myself on foot/by car | 17 [b] | −13 [a] | 6 [b] |
| A3 | I order online | 8 [b] | −10 [a] | 10 [b] |
| A4 | Brought by a friend/family member | 9 [b] | −12 [a] | 7 [b] |
| B1 | Hypermarket | 0 | 1 | 5 |
| B2 | Supermarket | 12 | −2 | 6 |
| B3 | Bakery | 3 | −3 | −1 |
| B4 | Convenience store | 0 | 0 | 2 |
| C1 | Product without packaging | −13 [a] | 3 [b] | 15 [b] |
| C2 | Product packaged in the store | −7 [a] | 8 [b] | 11 [b] |
| C3 | Pre-packaged product | −17 [a] | 6 [b] | 10 [b] |
| C4 | Frozen product | −17 [a] | −4 [b] | 7 [b] |
| D1 | No protective measures in the store | −3 [b] | 9 [b] | −19 [a] |
| D2 | Mandatory wearing of masks and distance keeping in the store | 1 [b] | 4 [b] | −12 [a] |
| D3 | Mandatory disinfection of the hands at the arrival at the store | 3 [b] | 6 [b] | −21 [a] |
| D4 | Limitation of the number of people in the store | 5 [b] | 9 [b] | −22 [a] |

Mindset 1:

*I go to the store myself on foot/by car*
*Supermarket*
*Product packaged in the store*
*Limitation of the number of people in the store*

Mindset 2:

*I order online*
*Hypermarket*
*Product packaged in the store*
*No protective measures in the store or limitation of the number of people in the store*

Mindset 3:

*I order online*
*Supermarket*
*Product without packaging*
*Mandatory wearing of masks and distance keeping in the store*

As the final part of the questionnaire, respondents were asked an optional open-ended question, which was "What is your opinion on the restrictive/protective measures currently in place?". The obtained results were supported by the short text responses to this question, some of which are shown in Table 5. Based on the comments, we concluded that people would have not supported the relaxations at the time of the completion of the study.

**Table 5.** The respondents' comments on the restrictive and protective measures, recorded at the end of the questionnaire.

| Mind-Sets | "What is your Opinion on the Restrictive/Protective Measures Currently in Place?" |
|---|---|
| MS-1 | Tightening for bakery products is required. Hungarian people have a bad habit of touching all non-packaged bakery products, fruits, and vegetables, so the restrictions and protective measures do not make much sense. They are pointless, except for hand disinfection. |
| MS-2 | It is better because there is no crowd. However, the mandatory 1.5 m distance is no longer considered by most of the people. I think the measures used in stores are good. The limitation of the number of the customers, the mandatory mask-wearing, and hand disinfection make me feel safer, but at the same time, public transport is still crowded. I agree with them basically. There are some things on which I would tighten up (e.g., smoking, eating on the streets) and I would like to see increased police presence. |
| MS-3 | I find the limitation of the number of customers a little unrealistic, since a lot of people can gather outside the stores, and they will be much closer to each other than inside. Of course, it is the large shops and malls that are the most important in this case. The measures are not feasible in the stores, and the stores do not comply with the regulations. It does not really matter, because it just slows down the spreading of the virus. Obviously, it is better than not having any measures, but the effective method would be mandatory vaccination. |

### 3.3. Response Time

BimiLeap® enables the researchers to conduct an evaluation of the response times of the participants. Response times are measured similarly to the ratings, e.g., each vignette receives a response time value, which is the time in seconds from the presentation of the vignette to the statement of the response (e.g., clicking on the rating). As these values have significant information content, the response times were also extracted and are presented in Table 6. Higher results mean a longer time needed to answer a vignette that contained

the given element. Elements that are selected quickly are the main factors in the creation of a "gut response", which is a reaction to a situation based on a person's instinct and feelings, rather than on a logical analysis. The highest values for total panel were registered for elements "Convenience store" and "Frozen product", meaning that participants required the longest time (e.g., it required a higher cognitive load) to answer vignettes containing these elements. The shortest times were recorded for the protective measures, meaning that these elements were easy to answer, and participants did not need a long time to formulate their ratings.

**Table 6.** Regression coefficients for response times of total panel and the three mindsets. Higher numbers mean slower reactions. The highest numbers of each group are colored as dark gray, while the lowest are colored by light gray.

| Code | Additive Constant | Total | Mind-Set 1 (*n* = 27) | Mind-Set 2 (*n* = 53) | Mind-Set 3 (*n* = 45) |
|------|------------------|-------|-----------------------|-----------------------|-----------------------|
| A1 | I go to the store myself, using public transport | 1.4 | 1.4 | 1.5 | 1.4 |
| A2 | I go to the store myself on foot/by car | 1.5 | 1.5 | 1.4 | 1.6 |
| A3 | I order online | 1.5 | 1.5 | 1.5 | 1.5 |
| A4 | Brought by a friend/family member | 1.6 | 1.7 | 1.4 | 1.6 |
| B1 | Hypermarket | 1.6 | 1.8 | 1.4 | 1.8 |
| B2 | Supermarket | 1.5 | 1.5 | 1.4 | 1.6 |
| B3 | Bakery | 1.6 | 1.5 | 1.6 | 1.7 |
| B4 | Convenience store | 1.7 | 1.9 | 1.6 | 1.6 |
| C1 | Product without packaging | 1.4 | 1.2 | 1.7 | 1.3 |
| C2 | Product packaged in the store | 1.5 | 1.1 | 1.6 | 1.5 |
| C3 | Pre-packaged product | 1.5 | 1.5 | 1.6 | 1.5 |
| C4 | Frozen product | 1.7 | 1.4 | 1.8 | 1.8 |
| D1 | No protection measures in the store | 1.4 | 1.6 | 1.4 | 1.1 |
| D2 | Mandatory wearing of masks and distance keeping in the store | 1.4 | 1.8 | 1.4 | 1.2 |
| D3 | Mandatory disinfection of the hands at the arrival at the store | 1.4 | 1.4 | 1.4 | 1.4 |
| D4 | Limitation of the number of people in the store | 1.4 | 1.7 | 1.4 | 1.3 |

Regarding mindsets, Mindset 1 had the shortest response times all related to packaging ("Product packaged in the store" and "Product without packaging", 1.1 and 1.2, respectively). Members of Mindset 1 rated these elements the fastest, while the longest time was needed to answer vignettes containing the elements "Hypermarket" or "Mandatory wearing of masks and distance keeping in the store".

Mindset 2 needed the longest times to rate the vignettes. Generally, the elements performed similarly (coefficients around 1.4) but "Product without packaging" and "Frozen product" were rated more slowly (1.8) compared to the other elements.

The highest range in the response times was observed for Mindset 3, where the fastest ratings were registered when elements "No protection measures in the store" or "Mandatory wearing of masks and distance keeping in the store" were presented (1.1 and 1.2, respectively), while "Hypermarket" and "Frozen product" required the longest time to answer.

From these results, the sharpest differences were observed between Mindsets 1 and 3, while Mindset 2 served as an in-between segment. These sharp differences were expressed mainly with elements about protective measures, which needed longer response times from Mindset 1 than 2.

## 4. Conclusions

Restrictions due to COVID-19 have been in force since March 2020 in Hungary, and the study was run at the peak of the pandemic's third wave. The obtained data showed that shopping routines have changed. Although many risk factors play a role in the transmission of the virus, different consumer groups can be defined based on what they consider the most unsafe. Such information can be used (1) to understand people's motives and (2) to define protective measures. In the presented study, we analyzed bakery goods,

but these results can be transferred to other goods as well, as the introduced methodology enables us to do so.

There were no significant differences among consumers' attitudes based on the traditional socio-demographic descriptors, which did not show significant differences among consumers; however, the mindset-based classification was able to identify the most discriminating elements. The three identified mindsets covered people who considered bakery products themselves as the highest risk (Mindset 1), people who considered the purchase method as the highest risk (Mindset 2), and people who considered being in the store as the highest risk of a potential infection (Mindset 3).

Further studies should be made to uncover how these events and restrictions have affected consumer behavior, in particular, how these patterns persist after the pandemic is over. To our knowledge, there has been no similar study conducted in the international literature. Comparisons among different cultures might give different results, since, in many countries, bakery products are sold as packaged products, which raises fewer food safety issues compared to the ones without any packaging.

**Author Contributions:** Conceptualization: B.B. and A.G.; methodology: B.B. and A.G.; software: B.B. and A.G.; validation: A.G.; formal analysis: B.B. and A.G.; writing—original draft preparation: B.B. and A.G.; writing—review and editing: B.B. and A.G.; funding acquisition: B.B. and A.G. All authors have read and agreed to the published version of the manuscript.

**Funding:** Supported by the ÚNKP-20-3-II-SZIE-23 New National Excellence Program of the Ministry for Innovation and Technology from the source of the National Research, Development and Innovation Fund. The Project was supported by the European Union and co-financed by the European Social Fund (grant agreement no. EFOP-3.6.3-VEKOP-16-2017-00005).

**Institutional Review Board Statement:** Ethical review and approval were waived for this study as research data has been robustly anonymized, such that the original providers of the data cannot be identified, directly or indirectly, by anyone.

**Informed Consent Statement:** Informed consent was obtained from all subjects involved in the study.

**Data Availability Statement:** The data presented in this study are available on request from the corresponding author.

**Acknowledgments:** B.B. thanks the support of the Doctoral School of Food Sciences, Hungarian University of Agriculture and Life Sciences. A.G. thanks the support of the Premium Postdoctoral Research Program of the Hungarian Academy of Sciences and the support of National Research, Development and Innovation Office of Hungary (OTKA, contracts No. K134260 and FK137577).

**Conflicts of Interest:** The authors declare no conflict of interest.

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
