# Peer review of "Purchasing Bakery Goods during COVID-19: A Mind Genomics Cartography of Hungarian Consumers"

_agronomy, doi:10.3390/agronomy11081645_

Round 1
Reviewer 1 Report
Writing style. Most fine. Needs a good edit for verb/noun agreement. Try to avoid the use of negative phrasing style. Reword as direct positive actions. Rephrase "without sparing" in the abstract.
Abstract. Volunteers is not an easily interpretable term. Refer to the type of sampling method using normal research terminology. Say snowballing since that is what you used later in the methods section.
Introduction/literature. The concern about food safety and the background provided on the pandemic are both logical and reasonable as justifications for the study on baked goods. However, the authors provide too much information about the pandemic crisis in Hungary that is unrelated to the consumption of unwrapped food products. It would be better to narrow the introduction to focus mostly on concerns regarding food consumption and food safety during a pandemic. Keeping line 95 in the paper, I would like to see more literature added here (or in a separate literature section) on the issue of other food safety studies that may have been conducted in the past. The authors could share much more about the consequences or risks of consuming or buying unpackaged foods in general (salad bars or food stalls/street food?) and also any prior work on baked goods.
Methods. The discussion of conjoint analysis or methods to study people's minds is better moved to the methodology section. In the introduction you need to make the argument for why studying baked goods sales during a pandemic is vital to the Hungarian economy or to some other intriguing purpose. I think the most appealing issue is the stability of the packaging construct in different settings/under different conditions.
Table 1 is used effectively to describe the survey protocol. However, lines 116 and 117 mention that literature was used to define the questions. The author should cite the literature in relation to each of the questions or provide this in table form so the reader may know the source of the test items. More literature should be in the introduction to make us understand that transportation, store, packaging style, and disease prevention measures in relation to food safety have been studied by others and why these are important details for creating a mindset picture of high risk/low risk reactions.
Some of the items designed are difficult because they are double-barreled. Going to the store on foot is quite different than going to the store by car. It is not clear why the authors combined these together – explain your thinking. Also mask wearing and social distancing are two different behaviors in the minds of consumers—what allows you to combine them here? In terms of place of purchase, perhaps put definitions in parentheses to clarify the important quality defines each of the types of places, e.g., how do hyper and super markets vary in ways that would affect food safety?
Were there any screening questions?
Regarding your vignette example I do not see question D represented in this example.
Results: Given the wide range of ages/high standard deviation and the importance of age in vaccinations and risk perception mindsets, perhaps reporting ages as class frequencies would be more interesting to get at generational differences.
Regarding the reported clusters, it is not clear what you said about finding 2 versus 3 mindsets, or having automatically calculated ones and ones that were re-analyzed manually – please clarify with respect to your tables, which is which?
Table 3 was a good idea for conveying complex information. Instead of bolding, just use the superscript letters a and b to denote significance.. There should be a note explaining what a and b mean - below the table. The explanation of table 3 needs to be improved. It is not clear what negative numbers represent versus positive numbers. It's hard to jump from the table to your finding statements. You have three things that concern the reader in interpreting your tables. (1) What it means that the number is positive or negative. (2) What it means that the number is high, medium or low and what is truly high versus truly low on this scale. (3) And what level of significance is showing up. As you discuss your results you should be teaching the reader to be able to go from the numbers in the tables to your statements about findings.
Table 4 needs a better explanation. Perhaps likening it to the way we interpret factor analysis would be helpful. It's hard to understand the difference between say a 12 in mindset one and a four or six in mindset two. Can you help us understand how significant these numbers are in describing the mindsets and in comparing the strength or validity of the ‘factor’ in each case? Also what do a and b stand for here - need a note below the table.
It was hard to get anything additional from table 5 and generally from the bottom score analysis. It was also difficult to understand how the bottom score analysis was different from the total score analysis. Maybe just delete this section and move on to the discussion of qualitative results
Regarding fast response time, are you trying to suggest that items that are selected quickly are more risky in the minds of respondents? Your inferences about response time speed need to be clearly explained along with your statistical results.
Conclusion. You suggest that different baked goods consumer groups can be defined based on what they consider the most unsafe. But what exactly should service providers do differently based on your research? Do you believe this will outlast the pandemic? And what other characteristics linked to the mindsets? Any time we find differences in outcomes we are most interested in who these different groups are.
Author Response
Response letter to the Reviewers
Purchasing bakery goods during COVID-19. A Mind Genomics cartography of Hungarian consumers
Barbara Biró1, Attila Gere1*
1 Hungarian University of Agriculture and Life Sciences, Institute of Food Science and Technology, Department of Postharvest, Supply Chain, Commerce and Sensory Science, H-1118 Budapest, Villányi út 29-43, Hungary
*Correspondence: gere.attila@uni-mate.hu, gereattilaphd@gmail.com
Reviewer 1
First of all, we would like to thank your detailed work and the helpful advice. Our research team has carefully read your comments and we answered them in detail. Our answers are listed below:
Writing style. Most fine. Needs a good edit for verb/noun agreement. Try to avoid the use of negative phrasing style. Reword as direct positive actions. Rephrase "without sparing" in the abstract.
- The phrase has been changed to “including” in the Abstract and the Introduction. Grammar has been corrected.
Abstract. Volunteers is not an easily interpretable term. Refer to the type of sampling method using normal research terminology. Say snowballing since that is what you used later in the methods section.
- The phrase has been changed to “participants”. Snowballing technique has been included.
Introduction/literature. The concern about food safety and the background provided on the pandemic are both logical and reasonable as justifications for the study on baked goods. However, the authors provide too much information about the pandemic crisis in Hungary that is unrelated to the consumption of unwrapped food products. It would be better to narrow the introduction to focus mostly on concerns regarding food consumption and food safety during a pandemic. Keeping line 95 in the paper, I would like to see more literature added here (or in a separate literature section) on the issue of other food safety studies that may have been conducted in the past. The authors could share much more about the consequences or risks of consuming or buying unpackaged foods in general (salad bars or food stalls/street food?) and also any prior work on baked goods.
- As the paper discusses consumers’ reactions to the pandemic and aims to map their ways of thinking, we believe that it is important to give proper background on the pandemic situation at the time of completing the study. Unfortunately, there is a lack of literature overview on the COVID situation of Hungary available in English, so we also wanted to fill this gap. Nevertheless, following the reviewer’s suggestion, we added some relevant papers to the introduction to highlight the food safety issues of unpackaged goods in general.
Methods. The discussion of conjoint analysis or methods to study people's minds is better moved to the methodology section. In the introduction you need to make the argument for why studying baked goods sales during a pandemic is vital to the Hungarian economy or to some other intriguing purpose. I think the most appealing issue is the stability of the packaging construct in different settings/under different conditions.
- The demonstration of the possible economic impacts is not the purpose of the current study; therefore, the topic was not mentioned in the introduction. The aim of this research to assess consumers’ thoughts and behaviour during the pandemic, in particular the perception of safety of food purchases. Bakery products were chosen because these products are purchased daily and therefore suitable as a 'model' food group.
Table 1 is used effectively to describe the survey protocol. However, lines 116 and 117 mention that literature was used to define the questions. The author should cite the literature in relation to each of the questions or provide this in table form so the reader may know the source of the test items. More literature should be in the introduction to make us understand that transportation, store, packaging style, and disease prevention measures in relation to food safety have been studied by others and why these are important details for creating a mindset picture of high risk/low risk reactions.
- Food safety related literature has been added. The questions and the elements were defined by a group of experts after a careful analysis of the bakery market of Hungary, and existing COVID-19 relevant literature (e.g. newspaper articles, Magyar Közlöny, recommendations of the Hungarian National Public Health Center (NPCH) and the World Health Organization (WHO). This sentence also has been added to the text.
Some of the items designed are difficult because they are double-barreled. Going to the store on foot is quite different than going to the store by car. It is not clear why the authors combined these together – explain your thinking. Also mask wearing and social distancing are two different behaviors in the minds of consumers—what allows you to combine them here? In terms of place of purchase, perhaps put definitions in parentheses to clarify the important quality defines each of the types of places, e.g., how do hyper and super markets vary in ways that would affect food safety?
- Going to the store on foot and by car have been combined, because these are not uses of public transport. People are usually driving/walking to the store alone, or with their families, relatives, therefore they are not in an enclosed space with strangers, like on public transport.
- Mandatory mask wearing and social distancing were the current combined regulations during the completion of the study, which is why they were put together in one element. Hyper and supermarkets are larger, more people are present at the same time, but they vary in their size and capacity.
- Supermarkets are located in both residential and city centre areas, their floor area is 400-2500 m2, they have thousands of stock keeping units (SKUs) and the number of their cash registers varies between 3-10 units. Hypermarkets are mainly located on the outskirts of cities, their floor area is over 2500 m2, they have more than 10.000 SKUs and the number of cash registers is more than 10. These definitions have been added to the footnote of Table 1.
Were there any screening questions?
- No, only the questions presented in the paper were used in order to get a random sample.
Regarding your vignette example I do not see question D represented in this example.
- Indeed, the example does not have an element from all four questions in order to provide a balanced presentation of the elements. This phenomenon is enumerated in the methods section: “Vignettes are combined using strictly one element from each silo, however, not all silos are used in order to create a balanced presentation of elements throughout the study.”
Results: Given the wide range of ages/high standard deviation and the importance of age in vaccinations and risk perception mindsets, perhaps reporting ages as class frequencies would be more interesting to get at generational differences.
- The key concept of Mind Genomics is that horizontal segmentation should be done based on what people think and not based on their socio-demographic characteristics. We do not argue with the importance of socio-demographic variables, but the method has been constructed to work on mind-set level, therefore we do not expect any significant differences between age groups. This is intended to be introduced by Table 3, where there are actually no differences between genders and places of residence. The lack of age variable is due to the limit of space, but it does not give any more information as the other variables.
Regarding the reported clusters, it is not clear what you said about finding 2 versus 3 mindsets, or having automatically calculated ones and ones that were re-analyzed manually – please clarify with respect to your tables, which is which?
- The methods section has been updated to clarify the clustering: “By default BimiLeap® clustered the respondents into two mind-sets, and then into three mind-sets, because they represent different patterns of thinking about the same topic [29]. Although, these analyses are done automatically by BimiLeap®, the presented study has been re-analysed manually to get full control over the data set. The applied clustering indices suggested that three mind-sets should be kept.”
Table 3 was a good idea for conveying complex information. Instead of bolding, just use the superscript letters a and b to denote significance. There should be a note explaining what a and b mean - below the table. The explanation of table 3 needs to be improved. It is not clear what negative numbers represent versus positive numbers. It's hard to jump from the table to your finding statements. You have three things that concern the reader in interpreting your tables. (1) What it means that the number is positive or negative. (2) What it means that the number is high, medium or low and what is truly high versus truly low on this scale. (3) And what level of significance is showing up. As you discuss your results you should be teaching the reader to be able to go from the numbers in the tables to your statements about findings.
- Bold format has been modified. Superscript letters denote the homogenous subsets defined by Tukey post hoc test, as it is mentioned in the title of the table. The level of significance is added to the title, as well as the explanation of the numbers.
Table 4 needs a better explanation. Perhaps likening it to the way we interpret factor analysis would be helpful. It's hard to understand the difference between say a 12 in mindset one and a four or six in mindset two. Can you help us understand how significant these numbers are in describing the mindsets and in comparing the strength or validity of the ‘factor’ in each case? Also, what do a and b stand for here - need a note below the table.
- As we work with regression coefficients, we cannot define exact ranges for high and/or low interest. Our research experience, however, suggests that any coefficients higher than 8 should be considered as high interest, or in this case, a strong feeling of safety. This is one of the reasons of using analysis of variance in order to determine if there is an existing significant difference between the mind-sets. If significant difference exists, we can say that the feeling of safety of one mindset is higher compared to the other(s). In order to define differences among three groups, we need to use a post hoc test. Superscript letters here also denote the homogenous subsets defined by Tukey post hoc test.
It was hard to get anything additional from table 5 and generally from the bottom score analysis. It was also difficult to understand how the bottom score analysis was different from the total score analysis. Maybe just delete this section and move on to the discussion of qualitative results.
- Table 5 and the section has been deleted.
Regarding fast response time, are you trying to suggest that items that are selected quickly are more risky in the minds of respondents? Your inferences about response time speed need to be clearly explained along with your statistical results.
- Elements that are selected quickly are the main factors in the creation of a “gut-response”, which is a reaction to a situation based on a person's instinct and feelings, rather than on a logical analysis. This sentence also has been added to the text.
Conclusion. You suggest that different baked goods consumer groups can be defined based on what they consider the most unsafe. But what exactly should service providers do differently based on your research? Do you believe this will outlast the pandemic? And what other characteristics linked to the mindsets? Any time we find differences in outcomes we are most interested in who these different groups are.
- Our results will help service providers to understand their customers. Information is provided about which optional protective measures should be introduce during the pandemic and which of these can be maintained after.
- We absolutely believe, it may have become part of everyday life, because people have become more epidemiologically aware, and got used to these measurements and advice.
- Since the groups created by Mind Genomics are not different from each other sociodemographically, so no such differences can be detected in this study.

Reviewer 2 Report
- This research used Mind Genomics . The traditional method for analyzing consumers’ preference is conjoint analysis. Can you provide the reasons that you chose Mind Genomics?
- What is the theoretical background behind this method? What is the meaning of coefficients of 16 different elements in Table 2? This is important for readers to understand the empirical results.
- How you transforming rate scales to Binarized rating in Table 2? Please show how to categorize the 16 elements in the OLS regression. Is it possible to replace OLS method by logit regression method for the estimations of coefficients of 16 elements?
- Please explained how can you test that coefficients are significantly different between two socioeconomic groups? Is total in Table 2 represents the result from total sample?
- In line 180-181, “Participants are then clustered using k-means clustering using these regression coefficients and Pearson 1-R distance to create similarly thinking clusters, so-called mind-sets.” Please explain this in detail for readers to understand the process of determining mind sets.
- Three groups of consumers are identified in this research. However, in the real world, how to find the people who is mind set 1, or mind set 2 or mind set 3? Please explained?
Author Response
Response letter to the Reviewers
Purchasing bakery goods during COVID-19. A Mind Genomics cartography of Hungarian consumers
Barbara Biró1, Attila Gere1*
1 Hungarian University of Agriculture and Life Sciences, Institute of Food Science and Technology, Department of Postharvest, Supply Chain, Commerce and Sensory Science, H-1118 Budapest, Villányi út 29-43, Hungary
*Correspondence: gere.attila@uni-mate.hu, gereattilaphd@gmail.com
Reviewer 2
First of all, we would like to thank your detailed work and the helpful advice. Our research team has carefully read your comments and we answered them in detail. Our answers are listed below:
- This research used Mind Genomics. The traditional method for analyzing consumers’ preference is conjoint analysis. Can you provide the reasons that you chose Mind Genomics?
- Mind Genomics has been used in many scientific and product development processes and is being used by multiple multinational companies which gives us a solid basis of the validity of the method. It approaches the problem through a unique way, through the mind of the respondents and gives us clusters of respondents who think similarly. This feature is unique and has not been used for such research.
- What is the theoretical background behind this method? What is the meaning of coefficients of 16 different elements in Table 2? This is important for readers to understand the empirical results.
- As regression coefficients describe the relationship between a predictor variable and the response, the extent of the obtained coefficients tells us the effect of the given element on the rating given by the participants. This sentence also has been added to the text.
- How you transforming rate scales to Binarized rating in Table 2? Please show how to categorize the 16 elements in the OLS regression. Is it possible to replace OLS method by logit regression method for the estimations of coefficients of 16 elements?
- “Rating is transformed by BimiLeap®. First, ratings of 1-6 on the scale are transformed to 0 (i.e., low / weak feeling), while ratings of 7-9 on the scale are transformed to 100 (e.g., high / strong feeling), creating the so-called top analysis. Then, a small random number is added to generate small standard deviation. That lack of variation would cause the OLS to crash. The OLS is performed on these data, elements (A1-D4) are used as independent variables and the binarized ratings are used as dependent variable (Table 2).”
- The reviewer is right, there are several other regression methods available to analyse the data, one of them is logit regression. Another option would be PLS regression, which should also be taken into account. Comparison of regression methods is also planned to be examined by our research group; however, this paper is not a methodological one. It is also important, that the developers of the method use OLS, which proved its validity over the years.
- Please explained how can you test that coefficients are significantly different between two socioeconomic groups? Is total in Table 2 represents the result from total sample?
- Individual regression models are run; therefore, each individual has a vector of sixteen elements (coefficients of the 16 elements of the study). As the individuals are classified in one of the sociodemographic groups, hypothesis testing can be run, which depends on the number of groups and the results of normality tests.
- Table 2 “Total” presents the results of the total sample n=128.
- In line 180-181, “Participants are then clustered using k-means clustering using these regression coefficients and Pearson 1-R distance to create similarly thinking clusters, so-called mind-sets.” Please explain this in detail for readers to understand the process of determining mind sets.
- The cited section is a detailed explanation of the clustering method. For further information see the following: Gere, A.; Moskowitz, H. Chapter 9: Assigning People to Empirically Uncovered Mind-sets: A New Horizon to Understand the Minds and Behaviors of People. In Consumer-based New Product Development for the Food Industry; Porretta, S., Moskowitz, H., Gere, A., Eds.; The Royal Society of Chemistry, 2021; pp. 132–149 ISBN 978-1-83916-139-1.
- Three groups of consumers are identified in this research. However, in the real world, how to find the people who is mind set 1, or mind set 2 or mind set 3? Please explained?
- It is possible to create a decision tree-based classification model, which is able to classify new respondents into existing mind-sets without the need of completing the whole BimiLeap® study. This method is called personal viewpoint identifier and aims to identify the elements which discriminate the mind-sets the most. Using these elements, the decision tree-based method can give the possible mind-set membership of the participants.

Round 2
Reviewer 2 Report
n.a.